# The Repressor C Protein, Pf4r, Controls Superinfection of *Pseudomonas aeruginosa* PAO1 by the Pf4 Filamentous Phage and Regulates Host Gene Expression

**DOI:** 10.3390/v13081614

**Published:** 2021-08-15

**Authors:** Muhammad Hafiz Ismail, Katharine A. Michie, Yu Fen Goh, Parisa Noorian, Staffan Kjelleberg, Iain G. Duggin, Diane McDougald, Scott A. Rice

**Affiliations:** 1Singapore Centre for Environmental Life Sciences Engineering, Singapore 637551, Singapore; muhd.hafiz@ntu.edu.sg (M.H.I.); YUFEN001@e.ntu.edu.sg (Y.F.G.); laskjelleberg@ntu.edu.sg (S.K.); Diane.McDougald@uts.edu.au (D.M.); 2The School of Biological Sciences, Nanyang Technological University, Singapore 637551, Singapore; 3Structural Biology Facility, Mark Wainwright Analytical Centre, The University of New South Wales, Sydney, NSW 2052, Australia; k.michie@unsw.edu.au; 4The iThree Institute, The University of Technology Sydney, Sydney, NSW 2007, Australia; parisa.noorian@uts.edu.au (P.N.); Iain.Duggin@uts.edu.au (I.G.D.)

**Keywords:** bacteriophage, gene regulation, biofilm

## Abstract

It has been shown that the filamentous phage, Pf4, plays an important role in biofilm development, stress tolerance, genetic variant formation and virulence in *Pseudomonas aeruginosa* PAO1. These behaviours are linked to the appearance of superinfective phage variants. Here, we have investigated the molecular mechanism of superinfection as well as how the Pf4 phage can control host gene expression to modulate host behaviours. Pf4 exists as a prophage in PAO1 and encodes a homologue of the P2 phage repressor C and was recently named Pf4r. Through a combination of molecular techniques, ChIPseq and transcriptomic analyses, we show a critical site in repressor C (Pf4r) where a mutation in the site, 788799A>G (Ser4Pro), causes Pf4r to lose its function as the immunity factor against reinfection by Pf4. X-ray crystal structure analysis shows that Pf4r forms symmetric homo-dimers homologous to the *E.coli* bacteriophage P2 RepC protein. A mutation, Pf4r*, associated with the superinfective Pf4r variant, found at the dimer interface, suggests dimer formation may be disrupted, which derepresses phage replication. This is supported by multi-angle light scattering (MALS) analysis, where the Pf4r* protein only forms monomers. The loss of dimerisation also explains the loss of Pf4r’s immunity function. Phenotypic assays showed that Pf4r increased LasB activity and was also associated with a slight increase in the percentage of morphotypic variants. ChIPseq and transcriptomic analyses suggest that Pf4r also likely functions as a transcriptional regulator for other host genes. Collectively, these data suggest the mechanism by which filamentous phages play such an important role in *P. aeruginosa* biofilm development.

## 1. Introduction

Bacteriophages are important factors in the diversification of bacterial species. In *Corynebacterium diphtheriae* and *Escherichia coli* O157, prophages, bacteriophage integrated into the host chromosome and are replicated as part of the host, have been shown to harbour genes that encode toxins and virulence factors, enhancing the pathogenicity of the host bacterium [1,2,3]. Endogenous oxidative stress in biofilms is also known to drive diversity in biofilm communities [4]. In *Salmonella*, a prophage-encoded superoxide dismutase was shown to be the primary oxidative stress defence mechanism for the bacterium [5], thus linking phage to oxidative stress responses in the bacterial host. Bacteriophage can also contribute to the regulation of host genes [6,7]; for example, a loss of virulence of *Ralstonia solanacearum* and *P. aeruginosa* was observed when infected by certain bacteriophages. Thus, prophages contribute to bacterial fitness and evolution through a combination of horizontal gene exchange, stress response and regulation of gene expression and behaviour through mechanisms such as transcription factors, sRNAs and even bacterial lysis [8,9].

Filamentous phage, Inoviruses, have recently been shown to be widespread in bacteria and archaea, suggesting a long evolutionary relationship with their hosts [10]. *P. aeruginosa* PAO1 carries two filamentous phage integrated into its genome, Pf4 and Pf6, found in tRNA genes at positions PA0729.1 and PA4673.1 [11,12]. While no phenotype has currently been ascribed to the Pf6 prophage, the Pf4 prophage has been shown to play a crucial role in biofilm development where it influences maturation, cell death, dispersal and variant formation of the biofilm [13,14,15,16,17]. These effects are seen where the Pf4 filamentous phage acquires a superinfective (SI) phenotype [11,13]. Normally, *P. aeruginosa* carrying a prophage confers resistance to plaque formation (infection) upon exposure to its own phage. However, when the phage acquires particular mutations, that mutant phage can reinfect the host, which we refer to here as superinfective [14,18]. Additionally, it was shown that a superinfecting Pf4 mutant was less virulent in a mouse model of lung infection, suggesting that the phage plays a role in mediating virulence in the host [13,19]. Unlike phage lambda, the Pf4 prophage continually produces phage particles without resulting in host cell death [20]. Recently, it was observed that SI phages produced by *P. aeruginosa* have mutations in a gene (located in the intergenic region of PA0716 and PA0717) that has homology to an immunity protein of the P2 and lambda phage called the repressor C protein [14]. Li et al. [21] demonstrated that repressor C, which they named Pf4r, was responsible for immunity to Pf4 infection and also regulated replication of the Pf4 phage by repressing the expression of *xisF4*. However, the specific mechanism by which the Pf4r controls superinfection remained unknown in the context of the mutated Pf4r* [14].

Additionally, it has been shown that transcription factors encoded by phage can regulate host gene expression. In enterohemorrhagic *E. coli* (EHEC), deletion of the bacteriophage regulatory genes, anti-terminators N and Q and lytic genes S and R decreased the expression of the locus of enterocyte effacement (LEE)-encoded type III secretion system. It was subsequently discovered that the bacteriophage-encoded Cro functioned as a transcriptional activator of LEE as well as host virulence genes responsible for fimbria and flagellar expression [22]. Given that deletion of Pf4 resulted in decreased virulence of *P. aeruginosa* PAO1 [13,19], it is possible that Pf4r also plays a role in the expression of host genes.

The mechanism by which Pf4r contributes to the appearance of superinfection was investigated here. We observed that a mutation at Ser4Pro in Pf4r* was responsible for loss of immunity against reinfection but not at Arg80Leu. Furthermore, structural and biochemical analysis of the Pf4r mutant reveals that the superinfective Pf4r protein is deficient in dimerisation, a structural requirement for effective DNA binding. This loss of dimerisation is likely to explain why the mutant Pf4r protein does not bind its native promoter site. Complementation of Pf4r to the Pf4-deficient mutant restored LasB activity to that of the wild-type *P. aeruginosa*. In addition, Pf4r contributed to an increase in genetic variation of *P. aeruginosa* and potentially contributes to generating diversity in biofilms, especially in the presence of endogenous oxidative stress. The potential role of Pf4r in regulating gene expression of other genes was also explored using ChIPseq and transcriptomic analysis.

## 2. Materials and Methods

### 2.1. Bacterial Strains and Culture Conditions

All bacterial strains (Appendix A) were maintained on Lysogeny Broth (LB) medium (BD Difco, NJ, USA), either in broth or on plates supplemented with 1.5% (*w*/*v*) agar when necessary. Bacteria were cultured in either LB broth or M9 minimal medium (48 mM Na_2_HPO_4_, 22 mM KH_2_PO_4_, 9 mM NaCl, 19 mM NH_4_Cl, 2 mM MgSO_4_ and 0.1 mM CaCl_2_, supplemented with 0.2% (*w*/*v*) casamino acids and 0.04% (*w*/*v*) glucose). For plasmid maintenance in *E. coli*, the medium was supplemented with either 100 μg/mL carbenicillin or gentamicin (Carb100 or Gm100). *P. aeruginosa* PAO1 strains carrying the pJN105 vector and its derivatives were grown in medium supplemented with 100 μg/mL gentamicin (LB Gm100 or M9 Gm100). The cultures were incubated overnight at 37 °C, at 200 rpm.

### 2.2. Genomic DNA and Plasmid DNA Extraction

Genomic DNA was extracted using the QIAamp^®^ DNA Mini Kit (Qiagen, Germany), while plasmid DNA was extracted with the FavorPrep plasmid extraction kit (Favorgen, Taiwan) as per the manufacturers’ instructions. The concentration was assessed using both a NanoDrop 2000 Spectrophotometer (Thermo Scientific, MA, USA) and Qubit^®^ 2.0 Fluorometer (Life Technologies, CA, USA).

### 2.3. Polymerase Chain Reaction

Polymerase chain reactions (PCR) were performed in a Mastercycler pro (Eppendorf, Germany) using either Platinum^TM^
*Taq* DNA polymerase (Invitrogen, MA, USA) or Q5 DNA polymerase (New England Biolabs, MA, USA) as per the manufacturers’ recommendation and as previously described [23]. The cycling conditions were: (1) initial denaturation at 95 °C for 3 min; (2) 25–30 cycles of 95 °C for 30 s, annealing at temperatures as specified (Appendix A) [24] for 30 s, and extension at 72 °C for the time as specified (Appendix A) [24] and (3) a final extension at 72 °C for 5 (Platinum^TM^
*Taq* DNA polymerase) or 2 min (Q5 DNA polymerase).

### 2.4. Gel Electrophoresis

Gel electrophoresis analyses were performed using either 1 or 2% (*w*/*v*) agarose dissolved in 1× TAE buffer (39.95 mM Tris-base, 11.42% (*v*/*v*) glacial acetic acid, 127.29 μM EDTA) [23]. One or two microliters of 6× DNA loading dye (Fermentas Life Sciences, MA, USA) was mixed with 10 µL of DNA or 1 µL of GeneRuler™ 1 Kbp or 100 bp DNA ladder (Fermentas Life Sciences, MA, USA) for loading into the gels and subsequent electrophoresis and image capture (Molecular Imager Gel Doc System, Bio-Rad Laboratories, CA, USA).

### 2.5. Site-Directed Mutagenesis (SDM) of Pf4r

The wild-type (WT) *pf4r* sequence was amplified with pf4r_ORF_FrontUp_F and pf4r_P_ORF_R primers (Appendix A) [24] using Platinum Taq DNA polymerase (Invitrogen, USA) and inserted into the pCR4 plasmid of the TOPO TA Cloning Kit (Invitrogen, USA) before it was transformed into TOP10 cells as per the manufacturer’s instructions. Site-directed mutagenesis (SDM) of the four *pf4r* nucleotide mutations on the resultant plasmid, pCR4_Pf4r, was carried out sequentially (Appendix A) using the Q5 Site-Directed Mutagenesis Kit (New England Biolabs, MA, USA) according to the manufacturer’s instructions and as previously described [23]. All plasmids were transformed into NEB5α cells through heat shock and grown overnight with constant shaking before being extracted. A plasmid with only the SNP 788826G>T was generated using the primers pf4r_SDM03_G-T_F and pf4r_SDM03_G-T_R to modify this region from pCR4_Pf4r to generate the plasmid pCR4_motifSNP.

### 2.6. Electrophoretic Mobility Shift Assay (EMSA) of Pf4r

The *pf4r* and mutated *pf4r* sequences were sent as targets to the Protein Production Platform (PPP) (Biopolis, Singapore) for expression and purification of C-terminal hexahistidine tagged (CT6His) proteins to be used for electrophoretic mobility shift assay (EMSA). Prior to EMSA, the proteins were diluted in 1X TE buffer (10 mM Tris-HCl pH 8.0, 1 mM EDTA). All primers used for amplification of EMSA targets were 5′ biotinylated (IDT, Singapore) (Appendix A) [24]. The *pf4r* target was amplified with pf4r_EMSA_F and pf4r_EMSA_R while the non-target was amplified with gapA_EMSA_F and gapA_EMSA_R using PAO1 genomic DNA as the template. The P*_pf4r_*_-doubleSNP_ probe, which contained the two SNPs in the promoter region, and the P*_pf4r_*_-G>T_ and P*_pf4r_*_-A>G_ probes, which contained an SNP in positions 1 and 2, respectively (Appendix A).

Protein–DNA complexes were separated from free DNA on 6% polyacrylamide DNA retardation gels (Invitrogen, MA, USA). All binding and chemiluminescence reactions were carried out with the Pierce LightShift Chemiluminescent EMSA kit (Thermo Scientific, MA, USA) according to manufacturer’s instruction. Binding reactions were performed with protein concentrations in lanes 1 to 10 at either 0 or 90 nM and 10 fmol of target probe for 30 min at 30 °C in Pf4r binding buffer (10 mM Tris, 1 mM DTT, 2.5% Glycerol, 5 µg/µL BSA, 0.05% NP-40, 80 mM KCl, 5 mM MgCl_2_ and 2 mM EDTA). After incubation, 5 µL of 5X Novex Hi-Density TBE Sample Buffer (Thermo Scientific, MA, USA) was mixed with the binding reaction. The protein–DNA complex was then electrophoresed for 120 min at 110 V in 0.5X TBE (40 mM Tris-Cl pH 8.3, 45 mM boric acid and 1 mM EDTA) buffer through a 6% polyacrylamide gel at 4 °C.

### 2.7. SEC-MALS Analysis of Purified Pf4r and Pf6r Proteins

SEC-MALS was performed according to previously established methods [25]. Briefly, Pf4r, Pf4r* and Pf6r samples (prepared in 20 mM HEPES pH 7.5, 300 mM NaCl, glycerol 10% (*v*/*v*), 2mM TCEP) were gel-filtrated on a Superdex 75 5/150 gel filtration column (GE Healthcare, IL, USA) equilibrated with PBS. The chromatography system was connected in-line to a miniDAWN light scattering unit (Wyatt Technology, CA, USA) and an Optilab T-rEX differential refractive index detector (Wyatt Technology, CA, USA). Loading volumes were 50 μL with protein concentrations of 20 mg/mL. Data were analysed in Astra 6 (Wyatt Technology, CA, USA).

### 2.8. Crystal Structure of the Pf4r* Protein

The crystallography of Pf4r* was performed using previously described methods [26]. For crystallization, an initial screening of Pf4r (29 mg/mL), Pf4r* (50 mg/mL) and Pf6r (17 mg/mL) proteins were performed in buffer containing 20 mM HEPES pH 7.5, 1 mM EDTA, 2 mM TCEP, 300 mM NaCl, 10% glycerol and 100 nL of precipitant solution from a number of commercial crystallisation screens in sitting drop trays. Crystals for both Pf4r* and Pf6r were isolated and optimised prior to data collection at the Australian Synchrotron on MX1. Crystals were frozen in precipitant solution with a 20–25% PEG200 cryoprotectant. Pf4r* crystals were soaked with a number of heavy atom derivatives and SAD data at a wavelength of 1.0055 Å was successfully collected on a Mersayl (mercury) soak. Crystal trays contained 50 mg/mL Pf4r*, 37% PEG3350, 50 mM Tris pH 7, 50 mM KCl and 5% glycerol. Phasing was performed in AUTOSOL [27] within the Phenix software package, and data refinement was performed in Phenix [28]. Building was carried out in Coot [29]. Pf4r* was built and refined at 2.7 Å using the SAD data. Once refinement was complete, this model was used for Molecular Replacement (using Phaser [30]) to solve the structures of Pf4r* at 1.99 A and Pf6r at 1.73 Å on crystals in the following conditions—50 mg/mL Pf4r*, PEGMME 550 25%, 50 mM HEPES pH 7.0, 10 mM MgCl_2_; and 17.7 mg/mL Pf6r, polyethene glycol monomethyl ether 5000 20%, ammonium nitrate 200 mM, 25% PEG 200. Analysis of the final models in PDBePISA [31,32] indicates that both proteins form various dimers within the crystal. See Appendix A for crystallographic data collection and refinement statistics.

### 2.9. Tubing Biofilm System for Phenotypic Assays and mRNA Analysis

Biofilms were allowed to develop for 2 d at room temperature in 15 cm long silicon tubing size 3.18 mm (Silastic^®^ laboratory tubing, Dow Corning, MI, USA) in a continuous flow reactor setup under sterile conditions. The biofilm system was fed with M9 minimal salts medium. During inoculation of the bacterial culture, the pump was switched off, and the upstream tubing was clamped to prevent backflow. Using a 3 mL syringe with a 26 G × 1½” needle, 2 mL of 1 × 10^8^ cells/mL bacterial culture was injected into the lumen of the tubing. The bacterial cells were allowed to attach to the tubing surface for 1 h at room temperature under static conditions. The tubing was then unclamped, and flow was restored at a flow rate of 9 mL/h.

### 2.10. Pyoverdine and Pyocyanin Production

For planktonic cultures, bacteria were inoculated in M9 minimal medium and incubated overnight at 37 °C, at 200 rpm. For biofilms, effluent was collected daily and filtered through a 0.2 µm filter. From the cell-free supernatant, pyoverdine was semi-quantified as previously described [33] by measuring the emission fluorescence (Ex: 400 nm, Em: 460 nm) while pyocyanin was semi-quantified by absorbance (OD_695_).

### 2.11. Production of Secreted Proteases

To determine levels of proteolytic activity, cell-free supernatant was added to elastin-congo red to measure LasB activity [34]. Briefly, cell-free supernatant was added to elastin–congo red dissolved in reaction buffer 1 (0.05 M Tris-HCl, 0.5 mM CaCl_2_, pH7.5). The elastin–congo red mixture was incubated at 37 °C for 2 h with shaking, and reaction stopped by adding EDTA to a final concentration of 10 mM. The mixture was pelleted, and supernatant was transferred to microtiter plate to measure absorbance at OD_495_.

### 2.12. Estimation of Replication Fidelity and H_2_O_2_-Induced Mutant Frequencies

Overnight cultures were diluted to OD_600_ 0.1 and allowed to grow to mid-log (OD_600_ 0.5). The cultures were washed with 0.9% NaCl twice before being treated with 25 mM and 50 mM of H_2_O_2_ for 30 min at 37 °C [35]. The treated cells were washed twice with 0.9% NaCl, serial diluted, spotted onto LB plates and incubated overnight at 37 °C to determine cell viability. To determine the H_2_O_2_-induced mutant frequency, 500 µL of treated and washed cells were inoculated into 4.5 mL of LB media and cultured overnight at 37 °C. Appropriate dilutions of the overnight cultures were then plated on LB plates and incubated overnight at 37 °C before enumerating the number of standard and variant colonies.

### 2.13. Construction and Expression of C-Terminal Hexahistidine Tagged Pf4r

Construction of plasmids and transformations were performed using standard methods as previously described [23]. The *pf4r* was amplified with pf4r_F_PstI and pf4r_R_CT6His_XbaI, and pf4r_SDM01_F_PstI and pf4r_R_CT6His_XbaI (Appendix A) [24] primers, respectively, and ligated into the *Pst*I-*Xba*I site of pBAD_yhjH, which has a pJN105 plasmid backbone, to generate the pPf4r and pPf4r* plasmid, respectively. The *yhjH* gene was excised with the digestion of pBAD-yhjh using *Pst*I and *Xba*I restriction endonucleases. The pPf4r and pPf4r* plasmids were transformed into *E. coli* TOP10 cells by heat shock, grown overnight and extracted. To generate pPf4r_A>G_ and pPf4r_C>A_, primers Pf4r_SDM1_AtoG_F and Pf4r_SDM1_AtoG_R and Pf4r_SDM2_CtoA_F and Pf4r_SDM2_CtoA_R were used with the Q5 Site-Directed Mutagenesis Kit (New England Biolabs, USA), respectively. The pJN105 plasmid backbone had a gentamicin resistance cassette. Therefore, to prevent selection of false-positive clones, the gentamicin resistance cassette in PAO1ΔPf4::Gm^r^ was excised. Briefly, the pFLP2 plasmid was electroporated into PAO1ΔPf4::Gm^r^ and spread on LB Carb100 plates after recovery. Transformants were then plated on LB agar with 10% sucrose (*w*/*v*) to counter select for the plasmid pFLP2. To ensure that the gentamicin resistance cassette was excised, transformants were then spread onto LB Gm100 plates to test for resistance. Transformants that could not grow on LB Gm100 plates were labelled as PAO1ΔPf4 for the insertion of extracted pPf4r, pPf4r_A>G_, pPf4r_C>A_ and pPf4r* plasmids. The empty pJN105 plasmid was transformed into *P. aeruginosa* PAO1 as a control to produce PAO1ΔPf4 pJN105.

### 2.14. In Vitro Immunity Assay of Pf4r Protein

Filtered phage effluent was obtained from overnight cultures of PAO1 by passing through a 0.2 µm Acrodisc syringe filter (Pall Life Sciences, NY, USA). Each filtered phage effluent was serially diluted for determination of Plaque Forming Units (PFU) using a modified top layer agar (TLA) method previously described by Webb et al. [36]. TLA plates were made by seeding the LB agar top layer with 5 mL LB soft agar (0.75% *w*/*v*) containing 750 µL of PAO1, PAO1∆Pf4 or PAO1ΔPf4 complemented with pPf4r, pPf4r_A>G_, pPf4r_C>A_ and pPf4r* M9 medium overnight culture. Ten microliters of each effluent dilution were spotted in triplicate onto TLA plates with PAO1, PAO1∆Pf4 and PAO1ΔPf4 complemented with pPf4r, pPf4r_A>G_, pPf4r_C>A_ and pPf4r* lawns. The plates were dried and incubated overnight at 37 °C. The PFU per mL for each sample was obtained from dilutions yielding plaque counts between 10 and 100 plaques per spot.

### 2.15. Quantification of pf4r Expression by qRT-PCR

The levels of gene expression of *pf4r* in PAO1, PAO1∆PF4, PAO1ΔPf4 pJN105 and PAO1ΔPf4 pPf4r was determined using qRT-PCR and the primers pf4r_qPCR_F and pf4r_qPCR_R (Appendix A) [23,24]. A standard curve was generated from the reactions carried out with serial dilutions of *pf4r* amplicon amplified using the same primers as the qPCR. Each qRT-PCR reaction mixture contained: 10 μL of 2X Fast SYBR^®^ Green Master Mix (Applied Biosystems, MA, USA) added to 0.2 μL of forward and reverse primers, respectively, and 1 μL of template DNA before being topped up to a final volume of 20 μL with deionised water. All qRT-PCR runs were performed using the StepOnePlus™ Systems (Life Technologies, CA, USA). The cycling conditions were: pre-cycling at 95 °C for 2 min, followed by 40 cycles of 95 °C for 10 s and annealing and extension at 60 °C for 30 s. Results were analysed using the StepOnePlus™ Software version 2.3 (Life Technologies, CA, USA), using the standard curve to quantify the amount of *pf4r* transcript in each sample.

### 2.16. Chromatin Immunoprecipitation Sequencing (ChIPseq) Sample Preparation

The ChIPseq protocol used here was modified from Bonocora and Wade [37]. Briefly, PAO1 pJN105, PAO1 pPf4r and PAO1 pPf4r* was inoculated in LB and incubated overnight at 37 °C, at 200 rpm in a rotary incubator. The following day, the overnight culture was diluted to a final OD_600_ value of 0.1 and a volume of 42 mL in a sterile flask. The diluted culture was incubated in a 37 °C rotary incubator to an OD_600_ of 0.3 before the addition of L-arabinose to a final concentration of 0.25% (*w*/*v*) with further incubation for 1 h. Molecular grade formaldehyde (Pierce Protein Biology, MA, USA) was then added to a final concentration of 1% and incubated at room temperature to crosslink any protein–DNA complexes. After incubation, glycine was added to final concentration of 0.5 M to neutralise any free formaldehyde in the medium. The culture was then pelleted and washed 3 times with 1 × TBS (50 mM Tris-Cl, pH 7.5, 150 mM NaCl) before finally being pelleted in a 1.5 mL tube. The pellet was resuspended in 1 mL FA lysis buffer (50 mM HEPES-KOH, 0.1% sodium deoxycholate, 0.1% SDS, 1 mM EDTA, 1% Triton-X100 and 150 mM NaCl) containing 4 mg/mL lysozyme and incubated for 30 min at 37 °C followed by 5 min on ice.

To completely lyse the cell pellet and shear the genomic DNA, the samples were sonicated using a VCX 750 probe sonicator (Sonics & Materials Inc., CT, USA) with the program of a 30 s on–off cycle for 30 min at 40% amplitude. After sonication, the lysate was transferred into a 1.5 mL tube on ice before being centrifuged for 5 min at 21,130× *g* at 4 °C to pellet cell debris. The supernatant was transferred to a 15 mL tube containing 2.5 mL prechilled FA lysis buffer and mixed by inversion.

The supernatant, containing the DNA–protein complexes, was then divided into three, 800 μL aliquots to which 20 μL of ChIP-Grade Protein A/G Magnetic beads (Pierce Protein Biology, MA, USA) and 20 μL of anti-His-tag Rabbit mAb (Cell Signalling Technology, MA, USA) were added. This mixture was incubated overnight on a Labquake^TM^ rotisserie (Thermo Fisher Scientific, MA, USA) at 4 °C. Protein A/G Magnetic beads and associated protein–DNA complexes were recovered by placing tubes on a magnetic rack for 2 min or until the supernatant had cleared. The supernatant was discarded and 700 μL FA lysis buffer was added and incubated for 3 min at room temperature on the Labquake rotisserie. This washing procedure was repeated with 2 rounds of 750 μL FA lysis buffer followed by 2 rounds of 750 μL 10 mM Tris-HCl, pH7.5.

The beads were recovered using the magnetic rack, and DNA–protein complexes were eluted by addition of 100 μL ChIP elution buffer (50 mM Tris-HCl, pH 7.5, 1% SDS and 10 mM EDTA) with incubation at 65 °C for 10 min with shaking at 500 rpm. The beads were removed by magnetic separation, and the supernatant was boiled to separate the DNA–protein complexes. DNA was recovered and purified using the Zymo ChIP DNA Clean & Concentrator (Zymo Research, CA, USA) as per the manufacturer’s instructions. The DNA was quantified with a Qubit Fluorometer (Thermo Fisher Scientific, MA, USA), and the quality was assessed with a High Sensitivity D1000 ScreenTape (Agilent, CA, USA) before the DNA was sequenced as 75 bp paired-end reads on an Illumina MiSeq platform.

### 2.17. ChIPseq Analysis

The quality of the ChIPseq reads was assessed using FastQC version 0.11.5 before it was adapter and quality trimmed using BBMap version 36.38 [38]. The data analysis was continued using the trimmed reads in R through the systemPipeR (version 1.12.0) package for pipeline control. The reads were aligned to the *P. aeruginosa* PAO1 genome using Bowtie2 version 2.2.6, and the output SAM file was converted to an indexed and sorted BAM file. In order to perform the peak calling, the MACS2 (version 2.1.1.20160309) [39] ‘callpeak’ function was used with the reads from *P. aeruginosa* pPf4r as the treatment file, and the *P. aeruginosa* pJN105 reads as the control. The identified peaks were annotated using the ChIPseeker package (version 1.14.2), and the sequences of the peaks were extracted using the GenomicRanges package (version 1.30.3) to facilitate motif discovery. The motif discovery was performed using the online tool MEME-ChIP [40], and the options used were set for an expected motif site distribution of ≥1 occurrence. Finally, the top-scoring motif was used to screen the *P. aeruginosa* genome to identify genes that may putatively be regulated by Pf4r using the online tool Virtual Footprint (version 3.0) [41].

### 2.18. Transcriptomic Analysis of Pf4r Complemented Biofilms

*P. aeruginosa* wild type, PAO1ΔPf4 and PAO1ΔPf4 pPf4r biofilms were formed for 2 d before the biomass was treated with RNAprotect Bacteria Reagent (Qiagen, Germany) and subsequently extracted using the RNeasy^®^ Mini Kit (Qiagen, Germany) inclusive of in-column DNA digestion. To ensure improved removal of contaminating DNA, the samples were further treated with TURBO™ DNase (Ambion^®^, TX, USA), and DNase removal and RNA clean up were performed using RNA Clean & Concentrator™ (Zymo Research, CA, USA) as per the manufacturers’ instruction. The RNA was converted into cDNA and sequenced as 100 bp paired-end reads using the Illumina HiSeq platform.

The total raw RNA reads were adapter and quality trimmed using BBMap version 36.38 [38] before the rRNA reads were depleted using SortMeRNA version 2.1 [42] to obtain mRNA reads. The mRNA reads were mapped using Bowtie2 against the *P. aeruginosa* PAO1 genome (NC_002516) modified to include the Pf6 prophage genome. HTSeq [43] was used to count the number of reads in each feature with the modified genome as a reference. Differentially expressed genes that were statistically significant were identified using the DESeq2 package [44] in R, and the datasets were tested using PERMANOVA to test for significant differences between the different conditions (strain and day). The log2 fold change values extracted at the end were based on the false discovery rate cutoff of 0.1.

## 3. Results

The effluent of biofilms formed by wild-type *P. aeruginosa* PAO1 and its isogenic, Pf4 deficient mutant (PAO1ΔPf4) were plated onto lawns of the wild type or the ∆Pf4 mutant. As previously observed, phage were detected on the PAO1ΔPf4 mutant lawn at all time points for the wild-type biofilm effluent (Figure 1). The plaque morphology observed is consistent with morphologies observed from previous studies of the Pf4 phage [13,36]. On day 4, we observed plaque formation on the wild-type lawn from the wild-type biofilm effluent, suggesting conversion to the superinfective (SI) form (Figure 1a). While the effluent from PAO1ΔPf4 initially produced no plaques, on days 6 and 7, similar numbers of plaques were observed on both the wild-type and PAO1ΔPf4 lawns (Figure 1c). Given that PAO1 carries the Pf6 phage in addition to the Pf4, we hypothesised that plaque formation from the PAO1ΔPf4 mutant was due to the conversion of the Pf6 into the SI form. Quantification of both Pf4 and Pf6 using phage specific primers showed the presence of both phage in the wild-type biofilm effluent and only the Pf6 phage in the PAO1ΔPf4 mutant (Figure 1b,d). For both the Pf4 and Pf6, there was an increase in phage numbers at the time the SI phage were observed on the lawns. 

### 3.1. The Structure of the Pf4r Protein

Both the Pf4 and Pf6 phage encode a small protein with homology to the P2 replication repressor, repressor C (RepC). We previously showed that the SI variant of Pf4 has mutations in Pf4r [45]. The mutated Pf4r, with non-synonymous mutations, will be referred to here as Pf4r*. Given that Pf4r shows homology to the P2 repressor C of *E. coli*, which functions as a DNA-binding protein, we speculated that Pf4r may similarly bind DNA to control the biofilm and lung infection phenotypes previously reported [14,21] and that those mutations might change the structure of Pf4r to alter its ability to bind DNA. We recombinantly over-expressed and purified Pf4r, Pf4r* (containing the mutations Ser4Pro and Arg80Leu) and a related repressor from the Pf6 phage (Pf6r), all with a C-terminal hexa-histidine tag for purification. We did not obtain crystals of the native Pf4r of sufficient quality to obtain a high-resolution structure of the protein. However, we were successful in generating high-quality crystals for the Pf4r* mutant from which the structure was directly determined and from which the Pf4r structure was modelled. The Pf4r* structure was solved initially by single anomalous dispersion (SAD) and then by molecular replacement (MR) to a final resolution of 1.97 Å in the space group C 2 2 21, yielding an asymmetric unit containing two protein chains (A and B). Each protein monomer is comprised of six alpha helices (Figure 2). The first three helices form a helix-turn-helix (HTH) domain. The crystal structure was assessed for biologically relevant dimerisation using the PDBePISA server [31,32], and two independent states of Pf4r* dimerisation were identified, Type A and Type B. Both are symmetric homodimers, each formed by the same 2-fold rotational axis, i.e., the type A homodimer is formed by two chain A monomer subunits rotated 180° to each other, and the Type B homodimer is formed by two chain B monomer subunits rotated around the same axis as Type A (Figure 2c). The complexes have Complex Formation Significance Scores (CSS) of 1.00 and surface areas of 763 Å^2^ and 733 Å^2^, respectively (as determined by PDBePISA), and a structural alignment to each other of RMSD 1.693 (determined using Super within Pymol [46]).

Pf4r shares only 27% amino acid identity with the distantly related phage P2 repressor C protein from *E. coli* (PDB 2XCJ). Nonetheless, the two symmetric Pf4r homodimer structures each have a homologous structure to the P2 repressor C (Figure 2b showing the Type A Pf4r* in brown superimposed on the P2 repressor C structure in cyan). The two proteins have very similar folds (RMSD 2.347 calculated using ‘Super’ within Pymol). The RMSD for an alignment of the Type B Pf4r* complex and the P2 repressor C is 2.625.

The two Helix 3s located at the bottom of the A-type complex of Pf4r* overlay closely with the equivalent Helix 3s from the P2-repressor C protein (Figure 2b). The H3 helices in the P2 repressor C are the DNA-binding helices that bind across two turns of the major groove, suggesting that Pf4r may also bind DNA in this fashion. In comparing the two different conformational variants of Pf4r* dimer structures (Type A and Type B) observed in the crystals (Figure 2c), we note two major differences between the two different complexes. In the Type B complex, the recombinant hexa-histidine tag is fully resolved, forming an extended loop at the top of the structure. This is of no biological importance and is not discussed further. The second major difference between the complexes is within the distances between the putative DNA-binding H3 helices (Figure 2b,c). Based on the sequence identity between Pf4r* and Pf4r, it is expected that Pf4r also binds DNA via these two helices, and the variation between the two complexes (Type A and Type B) suggests that there is some plasticity within the loops that orient DNA-binding helices. Similar to the P2 repressor C, the Pf4r binds to a non-palindromic DNA site, further accentuating the structural similarities between the two distinct proteins. Based on these similarities, we propose that for both Pf4r and the P2 repressor C, the plasticity of the helix placement contributes to the binding of non-identical DNA sites.

Crystals of the related wild-type Pf6r (1.73 Å) (51% identity to Pf4r) had three complexes in the asymmetric unit cell; however, structural alignments of these complexes showed almost no differences between each other (pairwise structural alignments of the three complexes calculated by Super Pymol range between an RMSD of 0.235 and 0.266) and the DNA-binding helices have the same spacing as that of Pf4r* Type A and the P2-repressor C protein (Appendix A).

### 3.2. Mutations in the Dimerisation Interface Causes Loss of Immunity Function

The two mutations isolated in Pf4r* were highlighted in the structure (Figure 3a). An Arg to Leu substitution at position 80 (Figure 3a, blue spheres) maps to an exposed surface of the protein on the opposite side of the dimer complex from the predicted DNA-binding motif and also well away from the dimerisation interface. We do not predict this mutation to affect DNA binding or dimerisation, but it is possible it may affect some protein–protein interaction with a currently unknown partner. Changing from Arg to Leu significantly alters the charge state of the surface. We note that in the P2 repressor C, a 9 amino acid deletion from the C-terminus (which would include the Arg/Leu 80) was fully functional, suggesting this region of the protein is not important for function [47].

The second mutation, a Ser to Pro mutation at residue 4 (78879A>G), was observed to reside directly at the dimer interface (Figure 3a, pink spheres). Aligning the Pf4r* structure with the closely related wild-type Pf6r structure (Figure 3b, bottom left), we can see that the proline mutation has not caused a large structural perturbation, i.e., the two structures (one wild type that dimerises and one being the mutant that does not dimerise in solution) (using SUPERPOSE v1.0) [48] are remarkably similar (RMSD 0.518 for backbone atoms) while being only 49% identical at the sequence level. Prolines typically break helices and lack hydrogen bonding potential on the α amino group and thus have reduced hydrogen-bonding potential. In the Pf4r* structure (Figure 3b), if we model a serine in place of the mutated proline to understand the wild-type case, there is a neighbouring aspartate residue across the binding interface on the opposite chain with which the serine residue is able to make two stabilising hydrogen bonds (Figure 3b, bottom right). The mutation in Pf4r* to proline at position 4 may mean that only weaker hydrophobic interactions can be made, and the homodimerisation may be less stable. While dimerisation was observed in the crystals, we believe dimerisation is an artefact of very high protein concentrations required for crystallisation.

It was already determined that Pf4r was responsible for conferring immunity to Pf4 infection [21]. However, the same was not determined for Pf6r. Despite our best efforts using multiple approaches, including that which was used to generate the PAO1ΔPf4 mutant, we could not delete the Pf6 prophage (data not shown). Hence, the rest of the study is focused on the Pf4 phage and Pf4r. To determine the role of the specific mutations in Pf4r* in the loss of immunity, plaque assays were conducted for wild-type PAO1, the mutant lacking the Pf4 phage (PAO1ΔPf4), as well as the phage deletion mutant that has been transformed with a plasmid carrying the wild-type *pf4r* or the mutations described above (PAO1ΔPf4 pPf4r, pPf4r_A>G_, pPf4r_C>A_ or pPf4r*). The ectopic expression of Pf4r and its mutants were induced using 0.25% (*w*/*v*) L-arabinose. As expected, supernatant from the wild-type *P. aeruginosa*-generated plaques (7.33 × 10^5^ PFU/mL) on the Pf4 deletion mutant and for the strain carrying the empty vector (PAO1ΔPf4 pJN105) (1.1 × 10^6^ PFU/mL), but did not infect the wild-type *P. aeruginosa* (Figure 3c). However, when Pf4r was expressed in the Pf4 deletion mutant, no plaque formation was observed. This suggests that Pf4r was sufficient to confer immunity against phage infection, in agreement with the work of Li et al. [21]. Unexpectedly, there was also no plaque formation for the complemented strain in the absence of arabinose induction. Therefore, *pf4r* expression was assessed by qRT-PCR (Appendix A). The copy number of *pf4r* was 7.22 × 10^7^ for arabinose-induced PAO1ΔPf4 pPf4r and 1.54 × 10^5^ without arabinose induction, much higher than was in the wild type (4.77 × 10^4^). Thus, the arabinose expression system is leaky, even in the absence of the inducer, evidently due to the difference in catabolite repression in *P. aeruginosa* as compared to *E. coli,* and this is in agreement with previously published work [49]. Therefore, for the remainder of this work, arabinose was not added to any of the cultures. In contrast, for the PAO1ΔPf4 strains complemented with the pPf4r containing mutation(s), there was a loss in immunity for the pPf4r_A>G_ (1.1 × 10^6^ PFU/mL) and the pPf4r* (1.15 × 10^6^ PFU/mL), while the pPf4r_C>A_ complementation remained immune. This loss of function could be due to the loss of dimerisation of the homodimer.

We expect that the serine to proline mutation (788799A>G) may affect the binding constant of the homo-dimerisation. Therefore, to determine if Pf4r forms dimers and the effect of this mutation on protein dimerisation, we performed gel-filtration size exclusion chromatography (SEC) and multi-angle light scattering (MALS). This revealed that both Pf4r and Pf6r form dimers in biological buffers. In contrast, Pf4r* eluted as a monomer with a molecular mass equivalent of a dimer as determined by MALS (Figure 4). This would suggest that the mutations in Pf4r* significantly reduce dimerisation and hence, prevent DNA binding.

### 3.3. Mutations in the Promoter Regions and in Pf4r Affects DNA–Protein Binding

The *pf4r* gene is localized in the intergenic region of PA0716 and PA0717 (788,542–788,808) [21,45]. Mutations were found in both the *pf4r* coding region (788570C>A and 788799A>G) as well as upstream of the ORF (788826G>T and 788857A>G) in SI isolates of *P. aeruginosa* [45] (Figure 5a). The position of the mutations in the 5′ region and as shown by Li et al. [21] suggests that mutations in the promoter region of *pf4r,* referred to here as P*_pf4r_* (to indicate the wild-type promoter), would affect Pf4r binding. The SI infective phage carried mutations in this region as well as non-synonymous mutations within the *pf4r* gene (Figure 5a), suggesting that SI could be due to the failure of the wild-type Pf4r to bind to the altered promoter region. Therefore, P*_pf4r_*_-G>T_, P*_pf4r_*_-A>G_ and P*_pf4r_*_-doubleSNP_ probes which contained the mutations in the putative promoter region previously observed [45] and the mutated protein, Pf4r*, were tested using in vitro binding assays (EMSA) (Figure 5b). As expected and confirmatory of what was demonstrated by Li et al. [21], the P*_pf4r_* probe showed slower migration after incubation with the wild-type Pf4r (Figure 5b-i) at 20 nM and showed two distinct bands that suggested the presence of both single (C1) and putative dimer or multimer protein–DNA complexes (C2). Only the dimer or multimer protein–DNA complex was observed at 80 nM of Pf4r. In contrast to the wild-type promoter sequence (P*_pf4r_*), the formation of a protein–DNA complex was observed only at ≥70 nM (Figure 5b-ii) for the P*_pf4r_*_-G>T_ probe, while there was no binding to both the P*_pf4r_*_-A>G_ and P_rC-doubleSNP_ probe even at 90 nM (Figure 5b-iii,iv). The mutated Pf4r* did not bind any of the DNA targets used (Figure 5b-vi–x).

In addition to regulating immunity against infection, we wondered if the Pf4r had additional roles in PAO1. Previous observations showed that the Pf4 phage was associated with increased virulence and biofilm formation by *P. aeruginosa* PAO1 [13]. This suggests that the Pf4r may also control host gene expression. We first assayed if Pf4r had an effect on the expression levels of some common virulence factors such as iron chelation as well as secreted proteases [50]. For this experiment, biofilms of wild-type *P. aeruginosa* PAO1, the Pf4 deletion mutant, as well as the Pf4 deletion mutant complemented with either pPf4r or pPf4r* were grown for 2 days, and their effluent was collected daily. In biofilms, Pf4r restored the LasB activity of the Pf4 deletion mutant to wild-type levels by the second day of biofilm growth (Figure 6c). The pyocyanin levels were lower in the Pf4 deletion mutant but were restored by both Pf4r and Pf4r* complementation on day 1, although there were not many differences on day 2 (Figure 6b). In the case of pyoverdine, the Pf4 deletion mutant overproduces pyoverdine on both days, but complementation with Pf4r and Pf4r* has no effect on pyoverdine levels (Figure 6c).

Mismatch repair and reactive oxygen and nitrogen species are linked to increased production of morphotypic variants, which are linked mutations in the host genome [14]. To determine if Pf4r contributed to an increase in variant formation, Pf4r and Pf4r* was complemented in both the ΔPf4 and the wild-type strains. There was no significant difference in the survival rates between the strains after 30 min of H_2_O_2_ treatment (Figure 6d). In contrast, while not a statistically significant difference, the percentage of variants induced by H_2_O_2_ was slightly higher when Pf4r was expressed in both the wild type and Pf4 mutant strains (e.g., 50 mM H_2_O_2_: 16.3% ± 2.5% for ΔPf4 pPf4r, 13.3% ± 1.6% for ΔPf4, 19.9% ± 0.7% for PAO1 pPf4r and 17.5% ± 2.3% for PAO1) (Figure 6e). This increase in variants was not observed when the Pf4r* construct was introduced into these strains. 

### 3.4. Pf4r Can Bind and Regulate Pseudomonas Genes

Based on the observation that Pf4r confers immunity against reinfection as well as affects LasB and pyoverdine expression as well as variant formation, we investigated whether Pf4r affects other *P. aeruginosa* genes. Therefore, to determine if Pf4r also functions more generally as a transcriptional regulator in *P. aeruginosa*, ChIPseq was performed with both Pf4r and Pf4r*. Based on the peaks detected by ChIPseq, a binding motif (SSRGGGCAAYADYTTCCTSGH) was predicted and subsequently used to scan the genome to find the potential regulon of the proteins.

No detectable DNA was obtained for the mutant Pf4r*. This is in agreement with the observation that Pf4r* did not bind any of the DNA targets (Figure 5b) due to the failure to dimerise (Figure 4). For the wild-type Pf4r, 46 peaks were detected, suggesting the protein binds to those genes (Figure 7a and Appendix A), with one of the binding sites being the P*_pf4r_* (i.e., as predicted based on the DNA binding data), here labelled as the intergenic region of PA0717 (peak region 787960–790399) (Figure 7a and Table 1).

To identify other potential regulons of Pf4r not detected by ChIPseq analysis, a putative binding motif was obtained using MEME-ChIP [40] from the sequences associated with the ChIPseq peaks. Four possible motifs were obtained, ranked by their E-value (Figure 7b). In the subsequent analysis, only the top-scoring motif, SSRGGGCAAYADYTTCCTSGH (motif 1), was used, which was found in 32 of the 46 regions identified in the ChIPseq dataset (Appendix A). 

An additional 853 potential binding regions were detected by Virtual Footprint using the top-scoring motif. Of those, 87 were located in intergenic regions and 766 within coding regions (Appendix A). A consensus table of ChIPseq peaks and Virtual Footprint matches was generated to assist with the analysis (Table 1). As expected, one of the genes detected by the Virtual Footprint analysis was the *pf4r* putative promoter region and was also detected as one of the ChIPseq peaks (Table 1, peaks 5a–c). Other genes observed in the ChIPseq dataset, *phzA1* and *pilQ*, which are associated with virulence, were also identified as part of the virtual regulon of Pf4r. The potential targets for regulation by the Pf4r include several other virulence-factor related genes such as *tle5b* (a T6SS effector) and *aprA* (a precursor of alkaline metalloproteinase), as well as genes associated with energy and central metabolisms such as *fixC* (an oxidoreductase) and *ambA* (a putative LysE-type translocator).

The ChIPseq analysis suggests that Pf4r controls *P. aeruginosa* PAO1 gene expression. To further explore the effect of Pf4r on *P. aeruginosa* gene expression, changes in global gene expression were determined. For this analysis, the wild-type *P. aeruginosa* and the deletion mutant complemented with the wild-type *pf4r* were compared relative to the Pf4 deletion mutant. As Pf4r* was not able to bind to any of the DNA targets during EMSA (Figure 5) and during DNA pulldown for ChIPseq, complementation with Pf4r* was not completed for RNAseq. As expected, the samples for the wild type and PAO1ΔPf4 mutant clustered separately, while PAO1ΔPf4 pPf4r also clustered away from the PAO1ΔPf4 mutant (Appendix A), suggesting altered expression by complementation of wild-type Pf4r.

Similarities in the differentially expressed genes (irrespective of whether they were induced or repressed) were compared between the Pf4r complemented strain as well as the wild-type *P. aeruginosa* strain relative to the Pf4 deletion mutant (Appendix A). This was to determine if the differences in gene expression of the wild type could be attributed to the presence of Pf4r to explain the reported phenotypic differences observed for the wild-type *P. aeruginosa* and the Pf4 deletion mutant (e.g., increased virulence and resistance to SDS) [13]. Several genes were upregulated across all of the comparisons, including PA2698 and PA1202, which are probable hydrolases, hypothetical protein PA2699 and *cdrA* (a cyclic diguanylate-regulated TPS partner). Genes that were identified in both the RNAseq and Virtual Footprint data sets (Appendix A) included *aprA*, *gapN* and *mexF*, while *cdrA* was also found in the ChIPseq data set as well as the RNAseq and Virtual Footprint data sets.

## 4. Discussion

The filamentous Pf4 phage has been shown to drive multiple biofilm phenotypes of its host, *P. aeruginosa*, including cell death, colony expansion, stability of the biofilm and morphotypic variant formation. The Pf4 phage is also associated with increased virulence in a mouse infection model [13], and it has recently been shown that there is a positive correlation between the presence of Pf phage and chronic lung infection [16]. These phenotypes are further linked to the conversion of the phage into a form that can reinfect the host, causing cell death. Here, we have shown through molecular methods and phenotypic assays how the Pf4 phage plays a role in aspects of the biofilm mode of growth through regulation of host gene expression, in addition to the previously demonstrated physical contribution of phage particles to the properties of the biofilm matrix [15,17].

### 4.1. Pf4r Binds to Conserved Promoter Sites as a Dimer

Pf4r shares homology with the *E. coli* phage P2 repressor C protein [17], and the P2 repressor C play a role in the lytic and lysogenic switch [51,52]. Central to these effects of the Pf4 phage in *P. aeruginosa* is the repressor protein, Pf4r, which not only controls immunity to infection and phage replication [21] but also acts to regulate the expression of host genes. Here, we demonstrated that the protein functions as a dimer to bind the target promoters.

The data are supportive data that the upstream region of Pf4r is the promoter of the *pf4r* gene as ChIPseq pulled down this region (Table 1). Additionally, the Pf4r binding motif was found in three sites of the region that overlapped with the direct and inverted repeats previously reported (Figure 7c) [21]. Mutations within this region were observed by Hui et al. [14] and McElroy et al. [45], and it was demonstrated here that the binding of Pf4r to that mutated promoter region was altered. It was also observed that there is a mutation in both binding sites of Pf4r, in the highly conserved guanine residue of the conserved motif (788826G>T, Figure 5a; motif site 5a, Figure 7c) and in a relatively conserved adenine residue (788857A>G, Figure 5a; motif site 5b, Figure 7c) [14,45]. The experimental data here showed that a single mutation could reduce Pf4r binding while mutations in the two sites completely abolished binding (Figure 5).

The Pf4r shows the closest homology to the repressor C protein of the temperate *E. coli* phage P2. The repressor C protein is a DNA-binding protein that forms a symmetric dimer, allowing it to bind non-palindromic direct DNA repeats via two helices (one located in each monomer) [47]. The helices are oriented in such a way that they can bind the major groove over two turns of the DNA. In support of this, the gel filtration and MALS result show that the Pf4r elutes as dimers and mutations arising in Pf4r* affect its ability to either fold or/and dimerise, causing its elution as a monomer. This would also explain why the mutant Pf4r* does not bind at all, even to the altered promoter region (Figure 5b), despite these mutations co-occurring in the superinfective Pf4-producing PAO1 morphotype [14,45]. The mutation in the dimerisation site (Figure 3a) also affected Pf4r’s function as an immunity protein as the 788799A>G (Ser4Pro) removed the host immunity to Pf4 infection while the 788570C>A (Arg80Leu) does not (Figure 3c).

### 4.2. Pf4r Mediated Regulation of Host Genes Associated with Biofilm Development, Virulence and Mediating Superinfection

Since Pf4r is a DNA-binding protein and is important for immunity against Pf4 reinfection, we hypothesised that complementing Pf4r in a Pf4 mutant would restore some of the strain’s phenotype to that of the wild type. Complementation was observed for LasB and pyocyanin activity in biofilms (Figure 6a–b). To assay for the global regulon affected by the Pf4r protein, ChIPseq and RNAseq were performed. Pf4r appears to regulate the expression of virulence factors, including *pldB* (Appendix A)*,* extracellular proteases and siderophores (Table 1, Appendix A). Based on the RNAseq data for the Pf4r-complemented PAO1∆Pf4, genes for intracellular proteases *clpP2* and *pfpI* were downregulated (Appendix A). These genes were previously shown to affect biofilm formation in *P. aeruginosa* [53]. While the results suggest that Pf4r plays a role in regulating the expression of these virulence factors, more work is needed to more clearly elucidate the mechanism of this regulation. For example, does Pf4r induce or repress virulence gene expression after binding to the promoter regions, or does Pf4r interact with the transcriptional regulators for these virulence genes?

It was further previously reported that biofilms of *P. aeruginosa* generate morphotypic variants at the same time the superinfective phage variant is observed [14] and that such variants are linked to mutations in the *P. aeruginosa* genome [33]. It is possible that in PAO1 biofilms, the appearance of variants is partly controlled Pf4r through repression of *pfpI,* which is involved in general stress protection [35]. Similarly, many of these morphotypic variants also produce SI Pf4 phages [14,45]. The SI Pf4 could cause the lysis of the wild-type strain, increasing the genetic diversity and overall fitness of the biofilm. Indeed, we have observed that overexpression of Pf4r, either in the phage deletion mutant or the wild-type *P. aeruginosa,* is associated with increased variant formation. It is therefore hypothesised that Pf4r represses expression of *pfpI*, which subsequently inhibits the general stress protection response, allowing for mutations to occur that lead to increased numbers of morphotypic variants as well as the superinfective variants. Further work is needed to explore the relationship between Pf4r, *pfpI* and the mechanism of variant formation. 

The data presented here as well as in previous studies [14,21,45] provide an insight into the possible mechanism of how the Pf4 phage is able to cause superinfection (SI) in *P. aeruginosa,* which is otherwise immune to Pf4 reinfection due to the presence of the Pf4 prophage within its genome. The normally expressed and released filamentous phage do not cause cell lysis. In the case of superinfection, it is possible that lysis is due to uncontrolled phage production, which overtaxes the membrane as the phage particles were released. As recently shown, an increase in Pf4 titres was due to a lack of repression of XisF4, which in turn positively regulates PA0727 (phage-initiator gene) [21]. This mechanism is similar to the *E. coli* λ phage, where mutations within the *cI* promoter disrupt repressor binding and induce the lytic life cycle of the phage [54,55]. As with Pf4, Pf6 also causes superinfection in *P. aeruginosa* but only from day 6 of biofilm growth. The plaques observed on day 6 are likely to be SI Pf6, based on the qPCR data; however, this could be further confirmed in the future by either purification and recovery of the phage or direct sequencing. As for Pf4, we hypothesise that similar mechanisms of gene regulation are involved for Pf6. However, due to the lack of a Pf6 deletion mutant, similar molecular and phenotypic assays could not be performed.

Filamentous phage clearly play important roles in other bacterial species, in particular in the context of association with a eukaryotic host. For example, the RSS1 phage of *Ralstonia* has been shown to increase bacterial virulence through increased polysaccharide production and virulence expression [56]. Other filamentous phage also have transcriptional repressors within their genomes. The cf phage of *Xanthomonas campestris* pv. *citri* has a hypothetical repressor protein that can interfere with RNA–RNA and DNA–protein interactions which regulate lysogeny [57]. In the *V. cholerae* CTXΦ phage, the RstR protein represses the *rstA* promoter by binding as a dimer-of-dimers and regulates phage replication [58,59]. Given that these filamentous phage have different regulatory genes, despite having otherwise conserved core genes, it is suggested that the regulatory functions have been acquired independently by each of the phage and hence may infer that they regulate immunity slightly differently and that they may also interact with the host in different ways, e.g., in terms of host gene regulation. As shown here, Pf4r plays a role in regulating a number of biofilm- and virulence-related genes, and those correlate with defects in biofilm development and virulence in vivo. Given the broader distribution of filamentous phage in bacteria and the conservation of the repressor protein [10,60], it is likely that filamentous phages are also important in regulating functions related to biofilm formation and host colonisation by those bacteria as well.

## 5. Conclusions

The data presented in this study suggest that the Pf4r is not only important in the control of superinfection but may also be an important regulator of virulence-factor-related genes. The Pf4r binds its putative promoter as a homodimer and regulates the expression of Pf4 genes. Mutations in the Pf4r at the dimer interface (788799A>G; Ser>Pro) prevented dimerisation from disrupting its function as a transcriptional regulator. The data here suggest that Pf4r contributes to the accumulation of mutations in the genome of PAO1, as well as mutations in Pf4r and its promoter region, converting Pf4 to its superinfective form. Given that filamentous phage have recently been reported in a broad range of bacteria and archaea from almost all habitats, it is possible that these phage play an important role in the regulation of biofilm and virulence responses of their hosts. This work highlights that simple genetic elements, such as the filamentous phage, can play a significant, albeit complex, role in the lifestyle and function of the host bacterium.

## Figures and Tables

**Figure 1 viruses-13-01614-f001:**
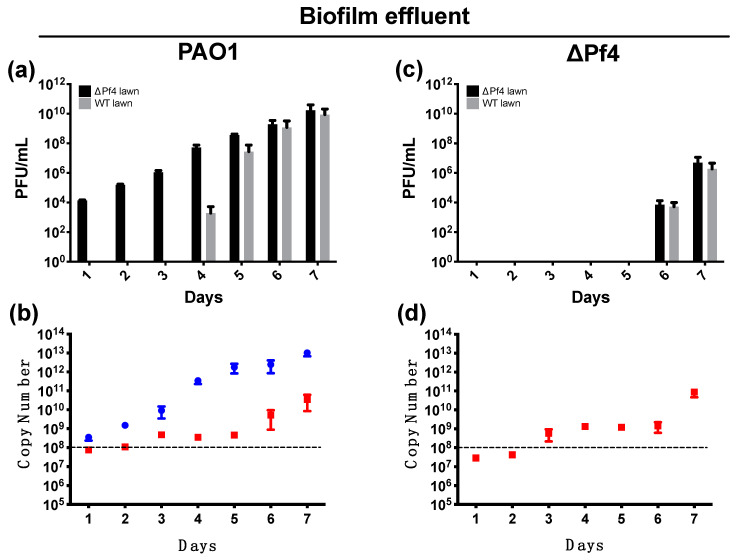
Phage production from biofilms formed by wild-type PAO1 (**a**,**b**) and the PAO1∆Pf4 mutant (**c**,**d**). The bars in black (■) represent the phage titre of the biofilm on a PAO1ΔPf4 top layer agar, while the bars in grey (■) represent the biofilm phage titre on a PAO1 top layer agar (**a**,**c**). The points in blue (●) and in red (■) indicate the amounts of Pf4 or Pf6 DNA, respectively, detected in the biofilm effluent (**c**,**d**). The amounts of phage DNA are represented as the calculated copy numbers and the limit of detection of 1 × 10^8^ (indicated by the dashed line, ---). The biofilms were developed over 7 d. The error bars represent standard deviations. For all figures, there were 3 biological replicates with 3 technical replicates for each, i.e., *N* = 3 and *n* = 3.

**Figure 2 viruses-13-01614-f002:**
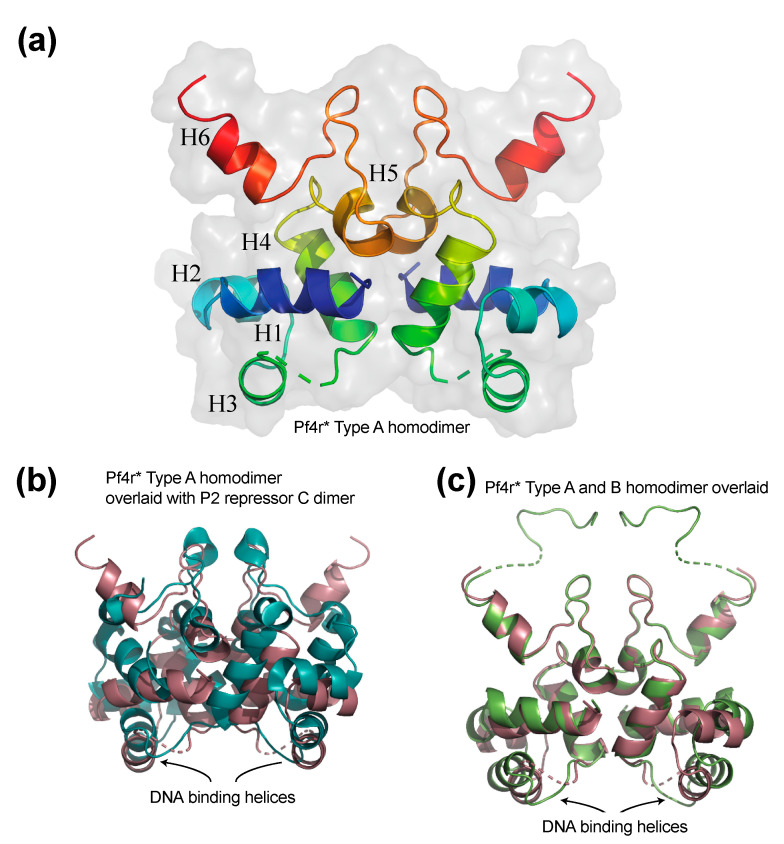
Crystal structure of Pf4r*. (**a**) The structure is coloured from the N-terminus (blue) to the C-terminus (red) in ribbon representation showing the symmetrical dimer formed by two chain As. The helices are numbered H1–H6, and the surface is rendered in transparent grey. (**b**) The Pf4r* Type A homodimer structure (brown) overlaid with the distantly related *E. coli* P2 repressor C (cyan) (PDB 2XCJ), showing high conservation in structure. (**c**) Overlay of the two types of Pf4r* symmetric homodimer (Type A in brown, Type B in green) observed in the solved crystal. Differences arise within the spacing of the predicted DNA-binding helices.

**Figure 3 viruses-13-01614-f003:**
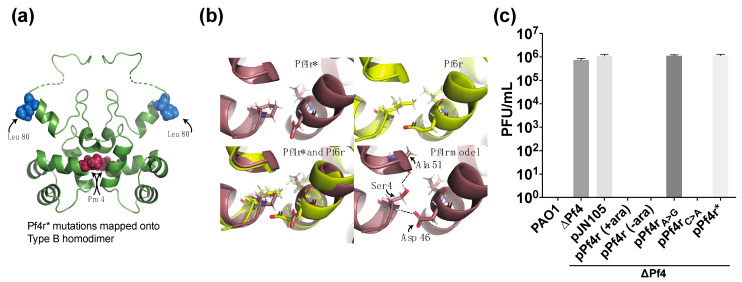
Mutation causing Ser4Pro causes loss of function as immunity factor. (**a**) Space-filling spheres for the two mutations in Pf4r* that stop DNA binding. Leu 80 (blue) is located on the top outside face of the complex and is unlikely to be involved in DNA binding or dimerisation. Proline 4 (pink) is located at the dimerisation interface and may perturb dimerisation. (**b**) Detailed view of the proline residue and the Asp 46 from the opposing dimer chain in close proximity in the Pf4r* crystal complex (top left) and the same region in Pf6r (top right). These sections are overlaid (bottom left). Bottom right shows that the wild-type Pf4r Ser4 residue modelled into the Pro4 position of Pf4r* may be able to make hydrogen bonds with the Asp46 on the opposing chain and form a stabilising interaction. Distances show hydrogen to oxygen distances in Angstrom for the two potential hydrogen bonds as determined in Pymol using the mutagenesis tool and measurement wizard. (**c**) Phage effluent from *P. aeruginosa* PAO1 was spotted onto different bacterial lawns, and the PFUs were enumerated. *N* = 3 and *n* = 3.

**Figure 4 viruses-13-01614-f004:**
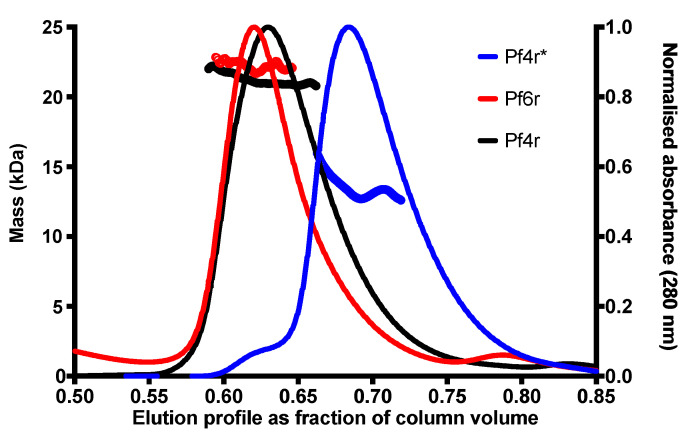
Pf4r (red), Pf4r* (blue) and Pf6r (black) elution profiles on SEC with molecular mass determination by MALS. Markers across the peaks represent the molar mass distribution (open circles, left axis, kDa) of the protein species within the peak as determined in-line MALS (Pf4r 21 kDa, Pf4r* 13.7 kDa and Pf6r 21.9 kDa). Theoretical masses of Pf4r and Pf4r* are 10.5 kDa (21 kDa dimer), and Pf6r is 11.2 kDa (dimer 22.4 kDa).

**Figure 5 viruses-13-01614-f005:**
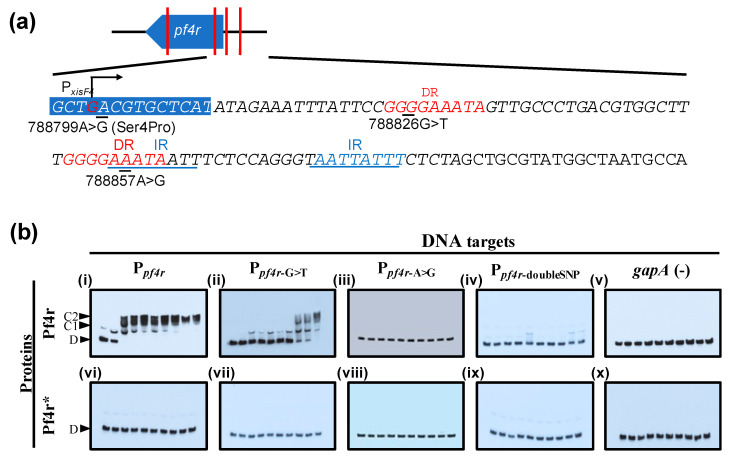
EMSA of Pf4r and Pf4r* protein gradients on target DNA. (**a**) The positions of *pf4r* relative to the Pf4 genes and the single nucleotide polymorphisms within the *pf4r* region relative to the PAO1 genome are indicated (red vertical lines). Italicised nucleotide sequences are part of the probe used in the EMSA. The locations of the direct repeats (DR), inverted repeats (IR) and promoter start sites were as seen in Li et al. [21]. (**b**) P*_pf4r_* indicates the wild-type promoter region. The P*_pf4r_*_-A>G_, P*_pf4r_*_-G>T_ and P*_pf4r_*_-doubleSNP_ probes have mutations in either 788857A>G or 788826G>T, or both positions, respectively. The binding of Pf4r and Pf4r* to the selected target regions was tested using an increasing protein gradient (lanes 1-10 were 0, 10, 20, 30, 40, 50, 60, 70, 80 and 90 nM, respectively). The non-target *gapA* (100 bp) gene was randomly chosen as the negative control for the assays. The notations indicate free DNA (D), protein–DNA complex (C1) and protein dimer–DNA complex (C2). Pf4r restores LasB activity to the Pf4 knockout and increases frequency of variants.

**Figure 6 viruses-13-01614-f006:**
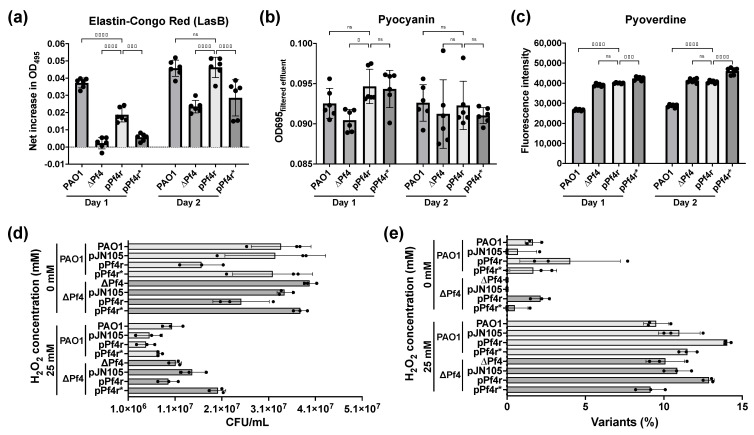
Pf4r restores LasB activity and slightly increases mutant frequencies. (**a**,**b**) Pyoverdine and pyocyanin production were semi-quantified by measuring emission fluorescence at 460 nm and by absorbance at OD_695_, respectively. (**c**) Extracellular protease activity in cell-free supernatant was measured using elastin-congo red for LasB (increase in absorbance at OD_495_). Significant difference was determined using Dunnett’s multiple comparisons test. (**d**) The number of viable cells after 25 mM H_2_O_2_ treatment for 30 min and (**e**) percentage of the mutant frequencies of ΔPf4 and PAO1 after treatment with H_2_O_2_. For all assays, *N* = 3.

**Figure 7 viruses-13-01614-f007:**
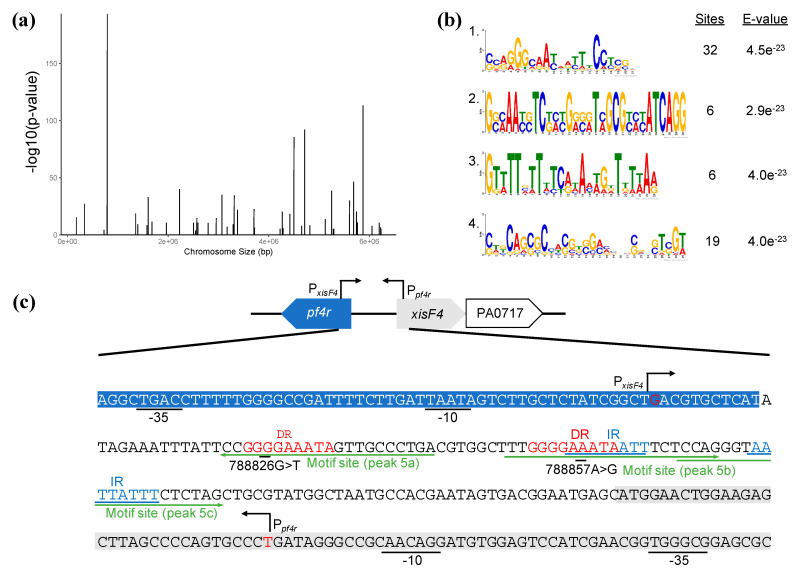
Peak coverage plot and predicted binding motifs of Pf4r. (**a**) The relative positions of the detected peaks bound by Pf4r in the PAO1 genome. The peaks’ heights were the -log10 (*p* value) obtained from MACS2 and represent the probability of the peak being a true match. (**b**) For every motif, the measurement of conservation is represented as a bit score (y-axis) for every nucleotide in their respective position (*x*-axis). The occurrence of the motif in the submitted sequences is represented as ‘Sites’. (**c**) Schematic diagram of the *pf4r* and *xisF4* intergenic region. The locations of the −10, −35, direct repeats (DR), inverted repeats (IR) and promoter start sites were as seen in Li et al. [21]. The Pf4r binding motifs as found by VirtualFootprint are underlined in green, found in the negative strand indicated by a left-facing arrow or in the positive strand when indicated by a right-facing arrow.

**Table 1 viruses-13-01614-t001:** Consensus ChIPseq peaks detected and predicted gene targets of the Pf4r.

Locus Tag	Gene	ChIPseq (MACS2)			Binding Motif (Virtual Footprint)					Gene Product
		Peak	Fold Enrichment	−log10 *p* Value	Conserved Motif	Strand	Distance from Gene Start	PWM	Location	
PA0154	*pcaG*	peak_1	2.50081	15.46022	CGAGGGCCACTATTACTTCCG	+	309	10.40	coding region	protocatechuate 3,4-dioxygenase, ⍺ subunit
PA0299	*spuC*	peak_2	3.20044	27.26002	CGAGGGCAACAAGATCCTCGA	+	-	10.68	coding region	polyamine:pyruvate transaminase
PA0717	PA0717	peak_5a	6.37126	179.65939	TCAGGGCAACTATTTCCCCGG	-	302	10.92	intergenic	hypothetical protein of bacteriophage Pf1
		peak_5b	1.9989	7.87121	TTGGGGAAATAATTTCTCCAG	+	274	9.87	intergenic	hypothetical protein of bacteriophage Pf1
		peak_5c	1.84109	6.05157	CCAGGGTAATTATTTCTCTAG	+	257	10.48	intergenic	hypothetical protein of bacteriophage Pf1
PA1249	*aprA*	peak_6	2.83107	18.79315	CCAGGCGAACAATTGCCCGCA	-	324	10.39	intergenic	alkaline metalloproteinase precursor
PA1450	PA1450	peak_8a	1.59375	3.78181	CAAGGGCAACGATTTCCTGGC	+	314	11.62	coding region	conserved hypothetical protein
		peak_8b	1.69336	4.73467						conserved hypothetical protein
PA1477	*ccmC*	peak_9	3.58171	33.07233	GCAGGGCAATAGCTTCCGCAT	+	-	10.90	coding region	heme exporter protein CcmC
PA1538	PA1538	peak_10	2.33997	11.48559	CCAGGGCAACCATTACACCGC	-	-	10.72	coding region	probable flavin-containing monooxygenase
PA1806	*fabI*	peak_11	2.28219	10.71977	GCAGGGCAAGTACAACCTCAC	-	206	9.79	coding region	NADH-dependent enoyl-ACP reductase
PA2035	PA2035	peak_12	3.95772	40.04172	TAGAGGAAATAGTTACATAGA	-	246	10.64	intergenic	probable decarboxylase
PA2306	*ambA*	peak_14	1.63948	3.92643	CCAGAGCAACAGTTGCCGAGA	+	57	9.78	intergenic	putative LysE-type translocator
PA2327	PA2327	peak_15	2.57107	14.75297	CGAAGGCAATAATGCCTTCCT	-	136	10.81	coding region	probable permease of ABC transporter
PA2345	PA2345	peak_16	2.28219	10.71977	CCAGGGCAACAGTTACCGGCG	-	108	10.73	intergenic	conserved hypothetical protein
PA2475	PA2475	peak_17	2.26436	10.55864	CCAGGGCAGCAGTTGCCCGGT	+	-	9.68	coding region	probable cytochrome P450
PA2612	*serS*	peak_18	2.57107	14.75297	CGGAGAGGATTATTTCCCCGA	-	246	9.98	coding region	seryl-tRNA synthetase
PA2715	PA2715	peak_19	3.72661	35.24215	GCAGGGAAATATTGCCCGGCT	-	193	10.29	coding region	probable ferredoxin
					GCCGGGCAATATTTCCCTGCC	+	192	11.07	coding region	
PA2953	*fixC*	peak_22	3.49856	34.68851	CGAGGGCAACTATATCATCTC	+	-	10.70	coding region	electron transfer flavoprotein-ubiquinone oxidoreductase
PA3316	PA3316	peak_25	1.83515	6.34533	CCGGGGAAATGATTTCCTACA	+	194	11.67	coding region	probable permease of ABC transporter
PA3766	PA3766	peak_26	1.80342	5.48047	CAAGGGCGAGAAGTTCATCAG	-	-	10.02	coding region	probable aromatic amino acid transporter
					TCATGACAACAACTCCCTGCA	+	-	9.77	coding region	
PA4021	PA4021	peak_30	5.40382	85.6179	CGAGGGCTGCTATTTCCTCCG	-	-	9.78	coding region	probable transcriptional regulator
PA4210	*phzA1*	peak_32a	5.29093	85.06443	CCGGAGAAACTTTTCCCTCGC	-	32	9.76	intergenic	probable phenazine biosynthesis protein
		peak_32b	1.9649	6.99041	CCGGAGAAACTTTTCCCTCGC					
PA4293	*pprA*	peak_33	2.07667	8.56197	CCGGGGCAACGTTTTCTCTGG	-	-	9.61	coding region	two-component sensor PprA
PA4576	PA4576	peak_34	2.45922	13.79064	CCAGGGTGATGGTTTCCTCGT	+	-	11.44	coding region	probable ATP-dependent protease
PA4704	*cbpA*	peak_37	1.78182	5.43821	GGGAGTAAATATCTTCCCGTG	-	73	9.73	intergenic	cAMP-binding protein A
					GCAGGCGGATCGGTTCTTCGT	-	-	9.64	coding region	
PA5040	*pilQ*	peak_39	3.86006	46.60398	GCAGGGCAATATCACCCTGCG	-	-	10.49	coding region	Type 4 fimbrial biogenesis outer membrane protein PilQ precursor
					GCAGGGTGATATTGCCCTGCA	+	-	10.3	coding region	
					CGAGGGCAACGTGGTCATCGA	-	-	9.91	coding region	
					GCGGGTCAAGCAGTTCCTGGA	-	295	9.84	coding region	
PA5089	*tle5b*	peak_40	2.82994	20.5095	GCGGGTGAATATCTGCCCCCA	+	-	9.82	coding region	type VI secretion phospholipase D effector Tle5b
PA5103	*puuR*	peak_41	2.26708	10.92837	GCAAGGAAATCGCTACCCCGT	+	273	10.44	coding region	probable transcriptional regulator
PA5205	PA5205	peak_42	6.52198	113.39552	GCGGGGTGATCTTTTCCTCGT	-	-	10.89	coding region	conserved hypothetical protein

## Data Availability

All sequencing data that support the findings of this study have been deposited in the NCBI Gene Expression Omnibus (GEO), accessible through the GEO series accession number GSE154459. The crystallographic data have been deposited in the Protein Data Bank (PDB) with the PDB IDs 6WNM and 6WPZ for Pf4r* and 6X6F for Pf6r. All other relevant data are available from the corresponding author upon request.

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
