# Peer review of "The Repressor C Protein, Pf4r, Controls Superinfection of Pseudomonas aeruginosa PAO1 by the Pf4 Filamentous Phage and Regulates Host Gene Expression"

_viruses, 2021, doi:10.3390/v13081614_

Round 1
Reviewer 1 Report
This work describes the structure and function of a filamentous phage repressor protein involved in superinfection exclusion. The authors present interesting data regarding the structure and DNA binding of the repressor, in particular of a defective mutant. However, the presentation of their data is poor and lacks controls. Also, there are multiple shortcomings that make their conclusions difficult to follow.
Major points
- Lines 340-344: Indeed, it is strange that you find plaques after 6 days in the deletion strain. To verify your interpretation you should clarify this point by sequencing a plaque and analyse whether it is indeed the Pf6 sequence responsible.
- 2 and section 360-380: This part of the manuscript should be rewritten for clarity. Obviously, the structure of Pf4r is missing and the lack of this control makes the section difficult to follow. Therefore, the authors should mention that they do not have the Pf4r structure and because of that they compare Pf4r* to P2-RepC. The heading of Fig. 2 “Crystal structure of Pf4r*” suggests that we see in (a) this structure, but we do not. This is confusing and has to be changed. Also confusing is, that the P2 repressor C is named inconsistently with RepC*, C-Repressor, P2 C-repressor, P2-C repressor etc.
Replace your wording: “…by two chain As”
- Section 400-408: This section should be rewritten for clarity. You switch between the Pf4r repressor A and B type back and forth. Mention that A and B are conformational variants of the same. Then, out of the blue, you have the sentence “Unusually, the P2 …” I guess this is because a structure of Pf4r is missing and then you fog in your conclusions using P2.
- Section 452-464: There are strange sentences, like “immunity to Pf infection” where you probably mean Pf4 and Pf6, or did you test Pf1, Pf3 etc? “Hence, the rest of the study…” Please rephrase! Also, the plasmid-controlled induction by arabinose should be mentioned early on when describing the plasmid.
- Line 470: copy number of what? The quantitation should be clearly described here or in the Fig.S3 legend. mRNA per cell? A standard reference control is missing here!
- 6 and section 525-548: I suggest that you move this section into the supplement. The data seem to show rather weak effects and, looking at your error bars, I have doubts that these are really significant.
- Likewise, the interpretation of the data regarding the altered expression of LasB and pfpI seems to overstate the rather weak effects.
- Line 656: This is described unclearly here. Shouldn’t it say: “This would also explain why Pf4r* does not bind at all and Pf4r not bind to the altered promoter…”
Minor points
Line 5: Some of the surnames are in capital letters. This should not be the case.
Line 23: Please repair: A mutation, Pf4r*, associated with…
Line 59: shown that a superinfecting Pf4 mutant…
Lines 79 and 80: Please avoid these details here and change to: …that a mutation at Ser4Pro in Pf4r*…but not at Arg80Leu.
Line 172: Tris pH 7
Line 217: remove the Error
Line 251: remove the Error
Line 252: remove the Error
Line 304: a space is missing at…. system PipeR
Line 321: Shouldn’t be: DNase
Line 378: two chain B
Line 416: “mapped onto” should be replace by “highlighted in the”
Line 494: The pf4r gene is localized in the intergenic region…
Line 497/8: Remove all but the first author
Line 505/6: Remove all but the first author
Line 519: Reference 18 is wrong. Maybe it should be 21.
Line 576: Reference 18 is wrong. Maybe it should be 21.
Line 640: Remove all but the first author
Lines 643/645: fix and remove the Errors. The description here is not understandable.
Line 648: At the end of the sentence refer to Figure 5.
Fig. 1 Replace the heading Efflux by “Biofilm efflux”
Fig. 2 please provide the color code for (b) and (c) in the legend
Fig. 3c Is this a biofilm effluent? If yes, after how many days?
Fig. 5a For clarity of the figure remove the arrow labeled with xisF (which is not mentioned)
This work describes the structure and function of a filamentous phage repressor protein involved in superinfection exclusion. The authors present interesting data regarding the structure and DNA binding of the repressor, in particular of a defective mutant. However, the presentation of their data is poor and lacks controls. Also, there are multiple shortcomings that make their conclusions difficult to follow.
Major points
- Lines 340-344: Indeed, it is strange that you find plaques after 6 days in the deletion strain. To verify your interpretation you should clarify this point by sequencing a plaque and analyse whether it is indeed the Pf6 sequence responsible.
- 2 and section 360-380: This part of the manuscript should be rewritten for clarity. Obviously, the structure of Pf4r is missing and the lack of this control makes the section difficult to follow. Therefore, the authors should mention that they do not have the Pf4r structure and because of that they compare Pf4r* to P2-RepC. The heading of Fig. 2 “Crystal structure of Pf4r*” suggests that we see in (a) this structure, but we do not. This is confusing and has to be changed. Also confusing is, that the P2 repressor C is named inconsistently with RepC*, C-Repressor, P2 C-repressor, P2-C repressor etc.
Replace your wording: “…by two chain As”
- Section 400-408: This section should be rewritten for clarity. You switch between the Pf4r repressor A and B type back and forth. Mention that A and B are conformational variants of the same. Then, out of the blue, you have the sentence “Unusually, the P2 …” I guess this is because a structure of Pf4r is missing and then you fog in your conclusions using P2.
- Section 452-464: There are strange sentences, like “immunity to Pf infection” where you probably mean Pf4 and Pf6, or did you test Pf1, Pf3 etc? “Hence, the rest of the study…” Please rephrase! Also, the plasmid-controlled induction by arabinose should be mentioned early on when describing the plasmid.
- Line 470: copy number of what? The quantitation should be clearly described here or in the Fig.S3 legend. mRNA per cell? A standard reference control is missing here!
- 6 and section 525-548: I suggest that you move this section into the supplement. The data seem to show rather weak effects and, looking at your error bars, I have doubts that these are really significant.
- Likewise, the interpretation of the data regarding the altered expression of LasB and pfpI seems to overstate the rather weak effects.
- Line 656: This is described unclearly here. Shouldn’t it say: “This would also explain why Pf4r* does not bind at all and Pf4r not bind to the altered promoter…”
Minor points
Line 5: Some of the surnames are in capital letters. This should not be the case.
Line 23: Please repair: A mutation, Pf4r*, associated with…
Line 59: shown that a superinfecting Pf4 mutant…
Lines 79 and 80: Please avoid these details here and change to: …that a mutation at Ser4Pro in Pf4r*…but not at Arg80Leu.
Line 172: Tris pH 7
Line 217: remove the Error
Line 251: remove the Error
Line 252: remove the Error
Line 304: a space is missing at…. system PipeR
Line 321: Shouldn’t be: DNase
Line 378: two chain B
Line 416: “mapped onto” should be replace by “highlighted in the”
Line 494: The pf4r gene is localized in the intergenic region…
Line 497/8: Remove all but the first author
Line 505/6: Remove all but the first author
Line 519: Reference 18 is wrong. Maybe it should be 21.
Line 576: Reference 18 is wrong. Maybe it should be 21.
Line 640: Remove all but the first author
Lines 643/645: fix and remove the Errors. The description here is not understandable.
Line 648: At the end of the sentence refer to Figure 5.
Fig. 1 Replace the heading Efflux by “Biofilm efflux”
Fig. 2 please provide the color code for (b) and (c) in the legend
Fig. 3c Is this a biofilm effluent? If yes, after how many days?
Fig. 5a For clarity of the figure remove the arrow labeled with xisF (which is not mentioned)
Reviewer 2 Report
This is a quite well-written, presented, and sound paper on the biology of certain filamentous phages affecting Pseudomonas aeruginosa and their ability to interfere with the virulence properties as far as biofilm formation and the like. Interesting aspects such as the relation of phage repressor proteins with classical ones such as the CI from lambda based on crystallized protein merits to be cited here.
Although more experiments could have been done, they would not justify at this moment a publication delay; hence I recommend publication in its present form.
Reviewer 3 Report
Dear Authors
Thank you very much for your manuscript submission. This manuscript represents invaluable and well-designed investigation. However, there are some revisions which should be done:
- Some parts of Materials and methods section including Bacterial strains and culture conditions, Genomic DNA and plasmid DNA extraction, Polymerase chain reaction, Gel electrophoresis, Site-directed mutagenesis (SDM) of pf4r, Electrophoretic mobility shift assay (EMSA) of Pf4r, SEC-MALS analysis of purified Pf4r and Pf6r proteins, etc. miss references. All sections should be referenced.
- The Primers used in this manuscript should be referenced.
- The References section suffers from old references. Refresh the references.
- Introduction and Discussion sections should be improved by refreshed references.
- This investigation is a part of Muhammad Hafiz ISMAIL's Doctoral thesis which should be mentioned in Acknowledgement section of the manuscript (Muhammad Hafiz Ismail. (2019). The role of bacteriophages in mixed microbial communities and populations of Pseudomonas aeruginosa. Doctoral thesis, Nanyang Technological University, Singapore) (https://dr.ntu.edu.sg/handle/10356/83268).
- Some of the used primers in this manuscript were designed by Muhammad Hafiz ISMAIL in his Doctoral thesis.
- The Muhammad Hafiz ISMAIL's Doctoral thesis should be mentioned as a reference in References section.
- Hence my decision regarding this manuscript is "Consider after major revision"
Round 2
Reviewer 1 Report
I have no further comments
Reviewer 3 Report
Dear Authors
Thank you very much for suitable revision. My decision is "Accept in present form"